# Kolmogorov–Arnold Graph Neural Networks

## Abstract

Graph neural networks (GNNs) excel in learning from network-like data but often lack interpretability, making their application challenging in domains requiring transparent decision-making. We propose the Kolmogorov–Arnold Network for Graphs (KANG), a novel GNN model leveraging spline-based activation functions on edges to enhance both accuracy and interpretability. Our experiments on five benchmark datasets demonstrate that KANG outperforms state-of-the-art GNN models in node classification, link prediction, and graph classification tasks. In addition to the improved accuracy, KANG's design inherently provides insights into the model's decision-making process, eliminating the need for post-hoc explainability techniques. This paper discusses the methodology, performance, and interpretability of KANG, highlighting its potential for applications in domains where interpretability is crucial.

## 1 Introduction

Neural networks and deep learning have clearly revolutionized countless fields, encompassing the domains of image, text, audio, and medical data. The basic building block of neural network models is the multilayer perceptron (MLP) (Haykin, 1998). Very recently, a promising alternative to MLPs was proposed, namely the Kolmogorov–Arnold Network (KAN) (Liu et al., 2024). MLPs are proved to be universal function approximators (Hornik et al., 1989), whereas KANs are inspired by the Kolmogorov–Arnold representation theorem (Kolmogorov, 1956; 1957), which states that a smooth multivariate function can be written as the finite sum of the composition of univariate functions; this has a similar structure to a two-layer neural network.

The idea of Liu et al. (2024) is to generalize this two-level structure to more levels. This extension allows the definition of a novel framework that proved increased accuracy and interpretability with respect to standard MLPs. Indeed, lack of interpretability in neural networks is a major issue, which limits its usage in domains in which interpretable results are crucial (e.g., medical and financial scenarios). To cope with this flaw, research focuses on the development of explainability techniques that try to unveil what was learned by those models in terms of salient features (Burkart & Huber, 2021; Saleem et al., 2022). In the domain of neural networks, models specialized in learning from graph-structure data have emerged. Graph neural networks (GNNs) (Scarselli et al., 2008) are able to leverage the connectivity structure of network-like data, learning representative topology-aware embeddings. In literature, we find different architectures addressing different aspects and characteristics of graph data. Graph convolutional networks (GCNs) (Kipf & Welling, 2017) extend the concept of convolution to graph structures, aggregating information from each node's neighbors using spectral graph convolutions. GraphSAGE (Hamilton et al., 2017) is another GNN model that generates node embeddings by sampling and aggregating information from neighbor nodes. It is based on an inductive learning procedure that allows embedding generation also for previously unseen nodes. Graph attention networks (GATs) (Velickovic et al., 2018) incorporate attention mechanisms into GNNs, allowing the model to focus on the most salient parts of a node's neighborhood. An additional GNN model is offered by GINs (graph isomorphism networks) (Xu et al., 2019). Such models are designed to have an effective discriminative ability, theoretically proven to be as powerful as the Weisfeiler–Lehman graph isomorphism test (Weisfeiler & Leman, 1968). GNNs have also been extended to relational structures, with the introduction of relational graph convolutional networks (RGCNs). They are designed to handle graphs with multiple types of edges and nodes by incorporating relations explicitly into the model. Furthermore, variational graph autoencoders (VGAE) (Kipf & Welling, 2016) aim to encode graph data into a latent space and then reconstruct the graph from the generated embeddings.

Similarly to classic multilayer perceptron models, GNNs are not immune to the interpretability curse; to shed light on their predictions, several explainability methodologies have been developed. Such methods explain GNN predictions in terms of importance subgraphs built of salient nodes, edges, node features, connected subgraphs, or a combination of those elements. The pioneering work in eXplainable Artificial Intelligence (XAI) for GNNs is GNNExplainer (Ying et al., 2019), which determined explanations by generating important subgraphs using a mask on the adjacency matrix able to maximize the mutual information between the prediction and the distribution of the possible explanation subgraphs. GraphSVX (Duval & Malliaros, 2021) in another method, which determines explanation in terms of important nodes and node features relying on the theoretical background behind Shapley values (Shapley, 1953). It uses a decomposition technique relying on a surrogate linear model for approximating Shapley values. A methodology targeting edges as a means for determining explanation subgraphs is EdgeSHAPer (Mastropietro et al., 2022). It employs Monte Carlo sampling to approximate Shapley values determining salient edges forming relevant subgraphs driving predictions. One additional XAI tool is SubgraphX (Yuan et al., 2021). Also exploiting Shapley value approximation, it looks for explanations only in terms of connected subgraphs by using a Monte Carlo Tree Search approach.

In this work, we extend the Kolmogorov–Arnold representation theorem to GNNs, introducing the Kolmogorov–Arnold Network for Graphs (KANG). Our main contributions are as follows:

- **Novel GNN Architecture:** We propose KANG, a novel GNN model that employs spline-based activation functions on graph edges. This design enhances the model's flexibility and interpretability while retaining the efficiency of message-passing mechanisms in GNNs.
- **Enhanced Interpretability:** KANG provides inherent interpretability by design, eliminating the need for external explainability techniques. This feature is crucial for applications in domains requiring transparent decision-making processes.
- **Performance Improvement:** We demonstrate that KANG outperforms state-of-the-art GNN models in node classification, link prediction, and graph classification tasks on benchmark datasets (Cora, PubMed, CiteSeer, MUTAG, and PROTEINS).

Recently, the integration of KANs with graph-structured data has attracted growing interest from researchers (Kiamari et al., 2024; Bresson et al., 2024; Zhang & Zhang, 2024). Specifically, Kiamari et al. (2024) proposed two KAN-based architectures: one where node embeddings are aggregated before applying the learnable spline-based KAN layers, and another where the KAN layers are applied prior to aggregation. They compared their models to GCNs using a reduced subset of the Cora dataset features (200 out of 1433). Bresson et al. (2024) introduced two GNN variants utilizing KAN layers for node representation updates: KAGIN (based on GIN) and KAGCN (based on GCN). Additionally, Ahmed & Sifat (2024) applied a similar architecture to molecular data for protein-ligand affinity prediction.

In the following sections, we detail the KANG[1] architecture and its components, present experimental results to validate our approach, and discuss the interpretability of KANG. Our findings suggest that KANG outperforms existing GNNs while providing interpretable outcomes.

## 2 METHODOLOGY

### 2.1 KOLMOGOROV–ARNOLD NETWORKS

This section details the construction and operation of our proposed KANG model. We begin by revisiting the key elements of Kolmogorov-Arnold Networks (KANs), upon which KANG is built. Kolmogorov–Arnold theorem states that a multivariate continuous function in a bounded domain can be rewritten using a finite composition of continuous functions on one single variable and the addition operation. Given $\mathbf{x}$ a vector of dimension $n$, $f$ a function such that $f : [0,1]^n \to \mathbb{R}$, it is thus possible to write

$$f(\mathbf{x}) = \sum_{q=1}^{2n+1} \Phi_q \left( \sum_{p=1}^{n} \phi_{q,p}(x_p) \right),\tag{1}$$

---

[1]For anonymization purposes the code of KANG is available here `https://anonymous.4open.science/r/KANGnn-2E0B/`

where $\phi_{q,p} : [0, 1] \to \mathbb{R}$ and $\Phi_q : \mathbb{R} \to \mathbb{R}$. Liu et al. (2024) extended Equation 1, representing a two-layer KAN with $2n + 1$ terms in the hidden layer, to larger depths and widths, parametrizing each one-dimensional function as a B-spline curve. This kind of neural network has an activation function on the edges instead of nodes; the latter simply perform a summation. From the implementation side, KANs activation functions $\phi(x)$ are built as a sum of a basis function $b(x)$ and the spline function such that

$$\phi(x) = w_b(x) + w_s \, \text{spline}(x). \tag{2}$$

The original KAN model uses

$$b(x) = \text{silu}(x) = \frac{x}{(1 + e^{-x})}, \, \text{spline}(x) = \sum_i c_i B_i(x). \tag{3}$$

At initialization, all activation functions are such that $w_s = 1$ and $\text{spline}(x) \approx 0$, and $w_b$ are initialized using Xavier initialization.

## 2.2 KOLMOGOROV–ARNOLD GRAPH NEURAL NETWORK

Building upon the strengths of KANs, we introduce the Kolmogorov–Arnold Network for Graphs (KANG), a novel GNN architecture (Figure 1) designed for processing graph-structured data. KANG leverages the flexibility, accuracy, and interpretability of KANs while retaining the efficient message passing mechanisms of GNNs.

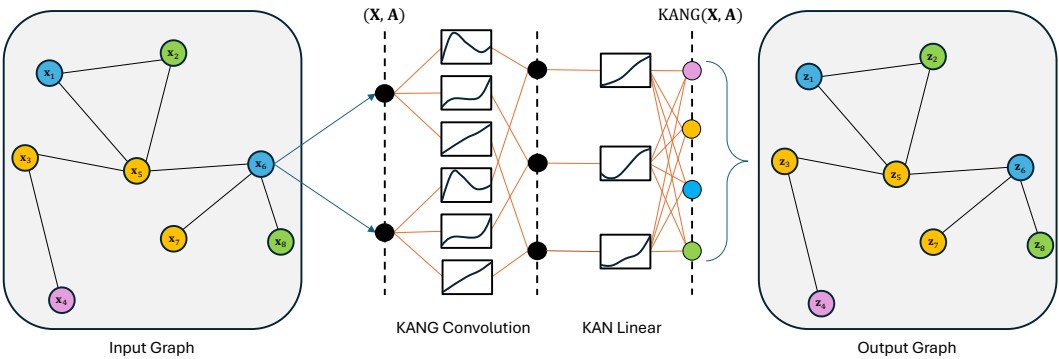

Figure 1: A simplified graphical representation of the KANG model is shown. $\mathbf{X}$ and $\mathbf{A}$ represent the feature matrix of the nodes and the adjacency matrix, respectively. The hidden layers consist of KANG convolutional layers, where messages are propagated and then aggregated. The output layer is a KAN linear layer. Each neuron has its own set of learnable splines. Although the figure provides a simplified view of the splines, the actual learned splines, responsible for transforming input values, can be visualized, as explained also in the original KAN paper (Appendix A.3). This allows for a clearer interpretation of the nonlinear transformations that contribute to the model's final predictions.

### 2.2.1 KANG ARCHITECTURE

KANG employs learnable spline-based activation functions on the edges of the graph, allowing for flexible nonlinear transformations of node features based on their connections. The architecture is composed of multiple layers:

- **KAN-based convolutional layer:** Each layer efficiently handles the propagation and aggregation of messages between nodes by applying a KAN-based transformation to the features of each node, taking into account information from its neighbors.
- **KAN-based linear layer:** A final linear layer performs a linear transformation on the aggregated node features, producing the final node representations.

The Xavier uniform initialization, also known as Glorot initialization, is a widely used method for initializing the weights of neural networks. This technique aims to maintain the variance of the

gradients approximately the same across all layers, thereby mitigating the vanishing and exploding gradient problems. The weights are initialized by sampling from a uniform distribution in the range $[-r, r]$, with $r = \sqrt{\frac{6}{n_{in}+n_{out}}}$, where $n_{in}$ and $n_{out}$ denote the number of input and output units in the weight tensor, respectively (Glorot & Bengio, 2010). As suggested in the KAN paper, also in KANG the basis function weights are initialized using the Xavier initialization, facilitating effective training and optimization.

### 2.2.2 MATHEMATICAL FORMULATION OF KANG

In the next part of this section we will go through the basic mathematical formulation of the constituent steps of KANG.

**Message Passing**: Each node $i$ in the graph has an initial feature vector $\mathbf{x}_i$. For each layer $l$ in the KANG, the node representations are updated through message passing and aggregation. The message from a node $j$ to its neighbors at layer $l$ is denoted as $\mathbf{m}_j^{(l)}$.

$$\mathbf{m}_j^{(l)} = \left[ \mathrm{spline}_1^{(l)}(x_{j,1}^{(l-1)}) \quad \ldots \quad \mathrm{spline}_H^{(l)}(x_{j,H}^{(l-1)}) \right]$$

**Spline-Based Activation Function**: The spline-based activation function $\varphi^{(l)}$ at layer $l$ used in KANG is defined as

$$\varphi^{(l)}(\mathbf{x}) = w_b^{(l)} b(\mathbf{x}) + w_s^{(l)} \mathrm{SPLINE}^{(l)}(\mathbf{x})$$

where $b(\cdot)$ is a basis function (e.g., SiLU), and $\mathrm{SPLINE}^{(l)}(\mathbf{x})$ applies $\mathrm{spline}_h^{(l)}(\cdot)$ to each element $h$ of vector $\mathbf{x}$.

**KANG Convolutional Layer**: Each KANG layer $l$ combines these steps, resulting in the following layer-wise update rule for node $i$:

$$\mathbf{x}_i^{(l)} = \mathrm{AGGR}_{j \in \mathcal{N}(i)} \varphi^{(l-1)}(\mathbf{x}_j^{(l-1)}),$$

where AGGR is aggregation function (we consider average, sum, max) which combines the messages that node $i$ receives from its neighbors[2]

**Output Layer**: After passing through multiple KANG convolutional layers, the final node representations are obtained using a KAN-based linear layer:

$$\mathbf{z}_i = \mathrm{KANLinear}(\mathbf{x}_i^{(L)})$$

where $L$ is the number of layers and $\mathrm{KANLinear}$ is a KAN layer as defined by Liu et al. (2024), which applies a final spline-based transformation.

### 2.2.3 OVERALL MODEL

The overall model can be summarized as:

1. Initialize spline weights with Xavier uniform initialization.

2. For each KANG layer $l = 1, \ldots, L$:

    (a) Compute messages $\mathbf{m}_j^{(l)}$ for each node.

    (b) Aggregate messages $\mathbf{a}_i^{(l)}$ for each neighboring node.

    (c) Update node representation $\mathbf{x}_i^{(l)}$ aggregating the spline-activated messages.

3. Apply a KAN-based linear layer to obtain the final node representation $\mathbf{z}_i$.

---

[2]Whereas the Kolmogorov–Arnold theorem suggests the use of summation as the aggregation function, we extend the concept by considering more general aggregation approaches, similarly to classical GNNs.

## 2.3 INTERPRETABILITY OF KANG

KANG allows for understanding its predictions without relying on external explainers, which may need to be trained (e.g., GNNExplainer) or whose computations are expensive (e.g., Shapley value-based explainers). KANG's interpretability involves determining the influence of input features and edge importance by analyzing the information flow across the graph.

### 2.3.1 FEATURE INFLUENCE

Motivated by previous studies (Baehrens et al., 2010; Simonyan et al., 2014; Hechtlinger, 2016; Sundararajan et al., 2017), to compute the incluence of features in KANG, we leverage the interaction between the gradients and the spline weights, capturing both feature sensitivity and nonlinear transformations. Specifically, let $\mathbf{x} \in \mathbb{R}^{N \times F}$ represent the input features for $N$ nodes, each with $F$ features. Given a hidden layer, the first step involves calculating the gradient of the output prediction with respect to each input feature, which gives us a matrix $\mathbf{G} \in \mathbb{R}^{N \times F}$, where each element $g_{i,f}$ reflects the sensitivity of the output for node $i$ to its corresponding feature $f$.

However, gradients alone do not provide a complete picture of how the features are processed by KANG, and can fail to determine a correct measure of feature importance (Sundararajan et al., 2017; Shrikumar et al., 2017). In our model, the spline function weights $\mathbf{S}_{\text{spline}} \in \mathbb{R}^{H \times F \times B}$, where $H$ is the number of hidden units at the current hidden layer, $F$ is the input feature size, and $B$ is the number of spline basis coefficients, modulate the feature transformation along the edges in a nonlinear fashion. These splines are critical because they allow KANG to adaptively adjust how features are processed based on the local graph structure, capturing important nonlinear interactions. We aggregate the spline weights by averaging over the $B$ coefficients, resulting in a reduced matrix $\mathbf{S}_{\text{mean}}^{(l)} \in \mathbb{R}^{H \times F}$, which can then be interpreted as a set of adaptive nonlinear weights acting on each feature across the hidden dimensions.

Next, to calculate the influence of feature $f$ of node $i$ on neuron $h$ at a given hidden layer $l$, we multiply the corresponding gradient $g_{i,f}$ by the mean spline weight $\mathbf{S}_{\text{mean}_{h,f}}^{(l)}$. This product reflects how a change in the input feature is transformed nonlinearly by the model's internal structure. The overall feature importance $\mathrm{I}_{i,f}^{(l)}$ for node $i$ al layer $l$ is computed by summing these contributions across all $H$ hidden units, producing a scalar value:

$$\mathrm{I}_{i,f}^{(l)} = \sum_{h=1}^{H} g_{i,f} \cdot \mathbf{S}_{\text{mean}_{h,f}}^{(l)}$$

This methodology integrates both the sensitivity of the features and the local transformations modeled by the splines, capturing the complex interactions between features that are fundamental to KANG's nonlinear structure. By doing so, we account not only for the direction of the change in prediction but also for the adaptive scaling that each feature undergoes along the graph edges.

### 2.3.2 EDGE IMPORTANCE

In KANG, edge importance captures how information flows between nodes, influencing predictions. By focusing on edge importance, we capture the interaction between nodes as modulated by the internal spline weights, which are central to the nonlinear transformations occurring along the edges. For a given node $i$, the importance of an edge $(i, j)$, where $j \in \mathcal{N}(i)$ is a direct neighbor of $i$, is determined by analyzing how the features of node $j$ are transformed and passed along the edge to node $i$.

This approach is particularly insightful because the spline weights $\mathbf{S}_{\text{spline}}$ play a crucial role in modulating the feature propagation along edges, allowing for adaptive nonlinear transformations of features as they flow through the graph. This makes the spline weights crucial for interpreting edge importance, as they capture complex feature interactions.

The spline weights in KANG serve a dual purpose: they not only determine how features are transformed between nodes but also govern the overall influence of edges on the model's prediction. When computing the influence of features on a node's prediction, the spline-modulated transformations provide insight into how individual features contribute to the target node. However, this

feature influence is inherently tied to the edge through which the features propagate. By extending this concept, we can derive the importance of an edge by aggregating the spline-modulated feature transformations and combining them with the feature activations. Essentially, edge importance emerges as a natural extension of feature influence, encapsulating how the transformation of each feature contributes to the overall prediction via the edge connecting two nodes.

Importances can be computed on a layer-by-layer basis. To determine the importance of an edge $(i, j)$ between a target node $i$ and a neighboring node $j \in \mathcal{N}(i)$, for the convolutional layer $l$, we first reduce the corresponding spline weights by averaging over the $B$ spline coefficients, yielding the reduced matrix $\mathbf{S}_{\text{mean}}^{(l)}$. This matrix encapsulates the non-linear transformation applied at layer $l$ as features propagate along the edges. We then incorporate the feature activations $\mathbf{a}_j^{(l)} \in \mathbb{R}^H$ and $\mathbf{a}_i^{(l)} \in \mathbb{R}^H$, where $H$ is the hidden dimension of layer $l$, representing the transformed feature vectors for the neighbor $j$ and the target node $i$ respectively, after layer $l$ of KANG. These activations reflect the local embeddings of the features post-convolution and are combined with the spline weights to assess the importance of edge $(i, j)$.

The edge importance, $\Xi_{i,j}^{(l)}$, for the edge $(i, j)$ at layer $l$ is computed by multiplying the spline-modulated feature weights $\mathbf{S}_{\text{mean}}^{(l)}$ with the weights $\mathbf{W}^{(l)}$ of the convolutional layer $l$, for which we are computing the importance. This product is then multiplied by the signals $\mathbf{a}_j^{(l)}$ and $\mathbf{a}_i^{(l)}$, summed over the hidden units $H$:

$$\Xi_{i,j}^{(l)} = \text{mean}(\mathbf{W}^{(l)} \cdot \mathbf{S}_{\text{mean}}^{(l)}) \sum_{h=1}^{H} \left( a_{h,i}^{(l)} \cdot a_{h,j}^{(l)} \right)$$

This formulation directly links nonlinear feature propagation along an edge to node $i$'s prediction. Since the spline weights remain static on the graph, they offer a consistent, interpretable framework for understanding how features are propagated through the edges, which are the fundamental pathways of information in a graph.

This approach to edge importance is highly beneficial because it breaks down the prediction process into localized interactions, helping us understand which edges are most responsible for driving the prediction. Unlike standard GNNs, where edges may simply aggregate features, KANG's spline-based transformations allow us to precisely identify how each edge modifies the feature representations in a nonlinear manner.

This section provided a comprehensive overview of the KANG architecture, its core components, and its interpretability capabilities. The next sections will delve into the experimental results and demonstrate the effectiveness of this new approach to GNNs.

## 3 EXPERIMENTS

### 3.1 DATASETS

We evaluated KANG on node classification, graph classification, and link prediction tasks using the benchmark datasets summarized in Appendix A.1. In the following section, we will discuss the experimental setup and the results obtained by KANG in comparison to state-of-the-art methodologies.

### 3.2 KANG PERFORMANCES

We compared the performances of our proposed model against established GNN architectures, namely GCN, GAT, using the GATv2 PyTorch Geometric implementation (Fey & Lenssen, 2019), GraphSAGE, and GIN. KANG was able to outperform all the mentioned GNN models in all tasks with all datasets, with the single exception of the link prediction with PubMed, in which GCN performed better.

Each method is unique, and the hyperparameters chosen for training are as crucial as the implementation itself. To ensure a fair comparison, each model has been trained following the guidelines

Table 1: Results in terms of average accuracy and standard deviation (%) on 10 runs (node and graph classification) and average AUC-ROC (link prediction) on the test set obtained by KANG and the compared architectures. Our framework delivers higher accuracy and ROC, being the top-performing architecture.

| Dataset | GCN | GAT | GraphSAGE | GIN | KANG (Ours) |
|---|---|---|---|---|---|
| **Node Classification** | | | | | |
| Cora | 77.5±1.0 | 78.7±1.1 | 73.6±2.5 | 75.5±1.2 | **79.5**±**0.8** |
| PubMed | 77.9±1.0 | 78.8±1.3 | 75.1±1.2 | 77.7±1.5 | **80.7**±**0.9** |
| CiteSeer | 67.6±1.5 | 68.7±0.5 | 63.1±2.3 | 63.1±1.9 | **69.1**±1.5 |
| **Link Prediction** | | | | | |
| Cora | 87.0±8.9 | 89.8±0.6 | 82.0±6.8 | 75.0±1.1 | **90.4**±0.5 |
| PubMed | **94.2**±0.4 | 88.9±1.1 | 84.8±3.7 | 89.5±0.5 | 85.8±0.4 |
| CiteSeer | 81.2±3.3 | 82.0±4.4 | 77.9±1.4 | 83.4±1.8 | **84.7**±0.6 |
| **Graph Classification** | | | | | |
| MUTAG | 67.5±4.0 | 64.0±5.4 | 74.0±8.3 | 74.5±6.5 | **93.0**±4.6 |
| PROTEINS | 71.3±2.4 | 71.0±2.3 | 71.4±2.5 | 72.4±1.2 | **73.7**±3.0 |

and hyperparameters provided by their authors. For training KANG, we conducted hyperparameter tuning to determine the best set of hyperparameter for each dataset and each task.

Hyperparameter tuning was conducted using a grid search over a broad range of potential values: learning rate [0.01, 0.001, 0.005], weight decay [1e-4, 1e-5], hidden channels [8, 16, 32, 64], dropout rate [0, 0.3, 0.6, 0.7], number of layers [1, 2, 3, 4], spline grid size [2, 3, 4, 8, 10, 20, 30], splines degree [1, 2, 4, 8, 10, 15], aggregation function [add, mean, max], and L2-regularization [0.0001, 0.00001]. All GNN models were trained for a maximum of 600 epochs for the Node Classification task, 700 epochs for Link Prediction, and 200 epochs for Graph Classification, with early stopping applied based on validation loss (node and graph classification) and validation AUC-ROC (link prediction). The optimal hyperparameters that we found for KANG are reported in Appendix A.2.

We report in Table 1 summarized results obtained by averaging over 10 runs the test accuray (or AUC-ROC for link prediction) of the models that achieved the highest validation accuracy during training over each run. For each run, to achieve unbiased outcomes, we randomly split the datasets utilizing 80% as training set, 10% as validation set, and the remaning 10% as test set.

Performance metrics alone do not fully capture a model's capabilities; it is equally important to evaluate its scalability to larger datasets. To assess how the models perform as graph sizes increase, we compared them on synthetically generated datasets under consistent conditions, ensuring all architectures had the same number of layers and hidden units, leading to comparable numbers of trainable parameters. The graphs varied in size (1000, 5000, 10,000, and 20,000 nodes) and edge density (with probabilities of 0.05, 0.25, and 0.5 for edge creation). Although KANG provides inherent interpretability (Section 2.3), this advantage comes with a slight increase in computational cost due to the additional parameters and weights introduced by the splines. As a result, the models were evaluated in two scenarios: 1) where only the training time is considered and 2) where all models were trained alongside an explainer, specifically GNNExplainer. KANG does not need an additional explainer, as it provides direct interpretability. The results of the study can be found in Appendix A.4, where we show that KANG can scale efficiently to large graphs.

## 3.3 INTERPRETABILITY

The added value of KANG lies not only in its higher accuracy but also in its inherent interpretability. As pointed out in Section 2.3, the interpretability of KANG is two-fold: it provides 1) a means for node feature influence and 2) a measure for edge importance, accounting for the information flow throughout the graph. As a representative example, we determined the most influencing features for a node in the Cora dataset, computed as shows in Section 2.3.1. This information can be used to understand the most important features for a particular node's prediction, crtitical in scenarios not suitable for black-box predictions (medicine, life sciences, and finance among others). We evaluated

the most influent features obtained in terms of Fidelity ($\text{FID}^+$) and Infidelity ($\text{FID}^-$) of prediction accuracy (Yuan et al., 2022), adapted for node classification.

The Fidelity metric is defined as

$$\text{FID}^+ = \frac{1}{N} \sum_{i=1}^{N} \left( \mathbb{1}\left( \hat{y}_i = y_i \right) - \mathbb{1}\left( \hat{y}_i^{1-m_i} = y_i \right) \right)$$

where $\hat{y}_i$ is the predicted label for node $i$ using the original graph with all the features and $y_i$ is its correct class, $\hat{y}_i^{1-m_i}$ is the predicted label for node $i$ using the graph with the $m_i$ most important features removed, $\mathbb{1}(\cdot, \cdot)$ is an indicator function and $N$ is the number of nodes for which the metric is computed. Analogously, the Infidelity metric is defined as

$$\text{FID}^- = \frac{1}{N} \sum_{i=1}^{N} \left( \mathbb{1}\left( \hat{y}_i = y_i \right) - \mathbb{1}\left( \hat{y}_i^{m_i} = y_i \right) \right)$$

where $\hat{y}_i^{m_i}$ is predicted lable for node $i$ when only the $m_i$ most important features retained. A good method should achieve high Fidelity and low Infidelity values. Table 2 shows the results considering different cutoffs for the top $k$ features.

Table 2: Feature influence analysis on the Cora dataset. The table presents the $\text{FID}^+$ and $\text{FID}^-$ scores for various top-$k$ feature cutoffs, computed on the correctly predicted samples from the test set. $\text{FID}^+$ and $\text{FID}^-$ values are stable across different cutoffs.

| Top $k$ | $\text{FID}^+$ | $\text{FID}^-$ |
|---------|------|------|
| 10% | 0.31 | 0.84 |
| 20% | 0.31 | 0.85 |
| 30% | 0.31 | 0.85 |

We notice how the removal of features for a single node leads to marginal changes in the prediction, identied by low $\text{FID}^+$ and high $\text{FID}^-$ scores, indicating that the behavior of GNNs is not solely dependent on node featues, but the overall structure and topology of the graph plays a crucial role. Indeed, the prediction of a node havily relies on the messages passed by its neighbors and not only on its own features.

After analyzing feature influence, it is also possible to interpret KANG to understand the importance of the edges in the graph. As shown in Section 2.3.2, is it possible to determine the importance of the information flowing on the egdes of the graph, in order to analyze the messages passed to a target node and determine the most influent neighbors impacting on the its prediction. We show a representative example on the Cora dataset in Figure 2, comparing the results against GNNExplainer.

In KANG, the values (which have been normalized for comparison) genuinely represent the network message, meaning they are the actual values used in making the prediction. In contrast, GN-NExplainer provides importance scores that are calculated post hoc on a subgraph optimized to maximize mutual information. We notice that both stategies prioritized similar edges. In particual, node with ID 2176 appears to be carrying the most important message for the target node 4 in both methodologies. Analogously, node 1761 is the least important neighbor (also beloning to a different class). This highlights that KANG interpretability is consistent with the explainability provided by GNNExplainer, validating the usage of the network messages as a means for edge importance. The added value brought by KANG is that it does not need an addtional explainer to be trained or used, thereby saving computation time and avoiding possible approximations introduced by such methodologies (Rudin, 2019).

Indeed, spline-based activation functions, which are central to KAN and consequently KANG, are inherently more interpretable compared to traditional neural network activation functions. Traditional GNNs use fixed, nonlinear activation functions like ReLU or Sigmoid, which can make it challenging to understand the decision boundaries or the transformations applied to the input features.

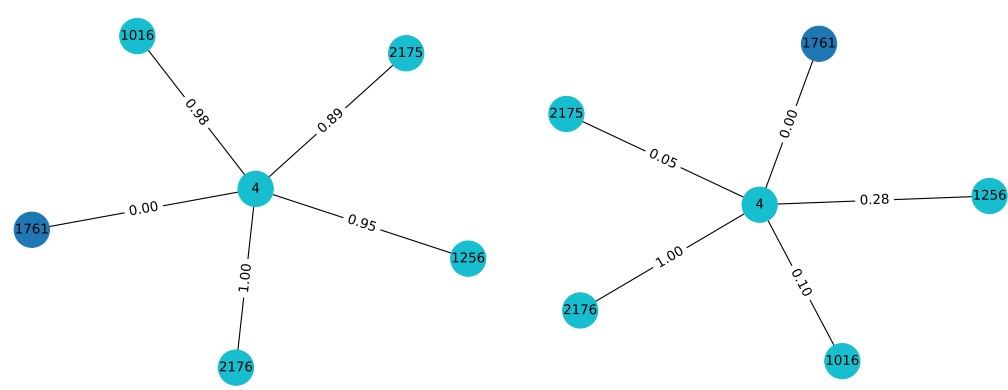

(a) Direct interpretation of KANG.    (b) Output of GNNExplainer applied to KANG.

Figure 2: Interpretability vs. explainability. Comparison of the direct interpretation of KANG (Figure 2a) and GNNExplainer applied to KANG (Figure 2b), trained on the Cora dataset for node classificaiton task. The explaination focuses on the node with ID 4. In Figure 2a, the output of the direct interpretation of the gradients and the weights of the splines of the neurons in the last convolutional layer (just before the KAN-based linear layer used for classification) is plotted. In Figure 2b, the edge mask returned by GNNExplainer is shown. The scores are normalized for visualization and comparison purposes. Additional examples can be found in Appendix A.5.

In contrast, spline functions are defined as piecewise polynomials which can be easily visualized and understood. The smooth and continuous nature of splines allows for a clear representation of how inputs are mapped to outputs. By examining the spline functions, one can see exactly how each input feature contributes to the final prediction and consequently how messages flowing throughout the graph influence the final output. This transparency makes it possible to trace the influence of individual features and edges, and understand the model's decision-making process.

## 4 CONCLUSIONS

In this paper, we propose KANG, a novel GNN architecture inspired by the Kolmogorov–Arnold theorem and based on the recently introduced KAN model. The added value of KANG is two-fold. First, it is more accurate than established state-of-the-art GNNs in node and graph classification and link prediction tasks. Second, thanks to the usage of splines and simple aggregation functions, KANG models are more interpretable. While KANG provides significant advancements in terms of interpretability, this is only a preliminary step. Future work should focus on enhancing the interpretability capabilities and exploring their application to even more complex graph structures, and compare the outcomes with state-of-art explainability strategies. Moreover, we showed that KANG can scale to large graphs.

We want highlight that KANG is not fully interpretable in every scenario, particularly in deeper networks where some information may be lost. However, KANG provides significantly greater transparency than other models by offering direct interpretability from the model itself, without relying on external explainability methods.

Looking forward, several research directions could further enhance KANG's performance and applicability. First, optimizing computational efficiency is essential, particularly by reducing memory usage and improving training and inference speed through techniques such as more efficient spline implementations, approximation methods, and parallelization. Second, incorporating edge features is a priority for extending KANG's capabilities. Developing methods to integrate edge information into the spline-based message passing and aggregation process could significantly improve predictive power.

Furthermore, applying KANG to real-world problems—such as biomedical research, where interpretability is crucial, or financial analytics, where it can aid in regulatory compliance—could demonstrate its practical utility and further validate the results with the aid of domain experts. Finally,

hybrid models combining KANG with advanced GNN architectures, such as attention mechanisms or variational techniques, may further boost performance while preserving interpretability.

By addressing these limitations and pursuing these research directions, KANG can be further extended into a more robust and versatile tool for graph machine learning applications.

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

# A APPENDIX

## A.1 DATASET DETAILS

Table 3 provides a summary of the benchmark datasets used in our experiments. The datasets were split into training, validation, and test sets with proportions of 80%, 10%, and 10%, respectively. The node classification and link prediction datasets include Cora, PubMed, and CiteSeer, while the graph classification datasets consist of MUTAG and PROTEINS.

Table 3: Summary of benchmark datasets. Datasets were split in train, validation, and test sets with proportions of 80%, 10%, and 10%, respectively.

| Node Classification and Link Prediction | | | | |
|---|---|---|---|---|
| Dataset | #nodes | #edges | #features | #classes |
| Cora | 2,708 | 10,556 | 1,433 | 7 |
| PubMed | 3,327 | 9,104 | 3,703 | 6 |
| CiteSeer | 19,717 | 88,648 | 500 | 3 |
| Graph Classification | | | | |
| | #graphs | #nodes | #edges | #features | #classes |
| MUTAG | 188 | ~17.9 | ~39.6 | 7 | 2 |
| PROTEINS | 1,113 | ~39.1 | ~145.6 | 3 | 2 |

## A.2 HYPERPARAMETERS

Table 4 presents the hyperparameters used for the various GNN models across the three main tasks: Node Classification, Link Prediction, and Graph Classification. The table includes the learning rate, weight decay, hidden units for each model, and the spline aggregation function used for KANG, along with dataset-specific variations for KANG.

Table 4: Summary of hyperparameters used to train the compared GNNs across the tasks of Node Classification (NC), Link Prediction (LP), and Graph Classification (GC).

| Task | Dataset | GNN | Hidden Units | Learning Rate | Weight Decay | Aggr. Fun. |
|---|---|---|---|---|---|---|
| NC | - | GCN | 32 | 0.01 | 5e-4 | - |
| | - | GAT | 64 | 0.005 | 6e-4 | - |
| | - | SAGE | 512 | 0.01 | 0 | - |
| | - | GIN | 32 | 0.01 | 0 | - |
| | Cora | KANG | 32 | 0.001 | 1e-4 | Mean |
| | PubMed | KANG | 24 | 0.001 | 1e-4 | Max |
| | CiteSeer | KANG | 64 | 0.001 | 1e-4 | Mean |
| LP | - | GCN | 32 | 0.01 | 5e-4 | - |
| | - | GAT | 64 | 0.005 | 6e-4 | - |
| | - | SAGE | 128 | 0.01 | 1e-4 | - |
| | - | GIN | 32 | 0.01 | 0 | - |
| | Cora | KANG | 64 | 0.001 | 6e-4 | Mean |
| | PubMed | KANG | 32 | 0.001 | 1e-4 | Mean |
| | CiteSeer | KANG | 64 | 0.001 | 1e-4 | Add |
| GC | - | GCN | 32 | 0.01 | 5e-4 | - |
| | - | GAT | 64 | 0.005 | 6e-4 | - |
| | - | SAGE | 512 | 0.02 | 0 | - |
| | - | GIN | 32 | 0.01 | 0 | - |
| | - | KANG | 32 | 0.01 | 1e-4 | Add |

## A.3 SPLINES VISUALIZATION

In the KANG architecture is it possible to visualize the splines learned during training. The spline weights from the last layer are extracted and plotted (Figure 3) based on the grid coordinates defined within the model. This allowed us to observe how the non-linear transformations learned by the model act on the input data before producing the final prediction. Visualizing the splines in the last layer provides useful insight into how the features are transformed before being mapped to the output space, offering clear understanding of how the model arrives at its predictions.

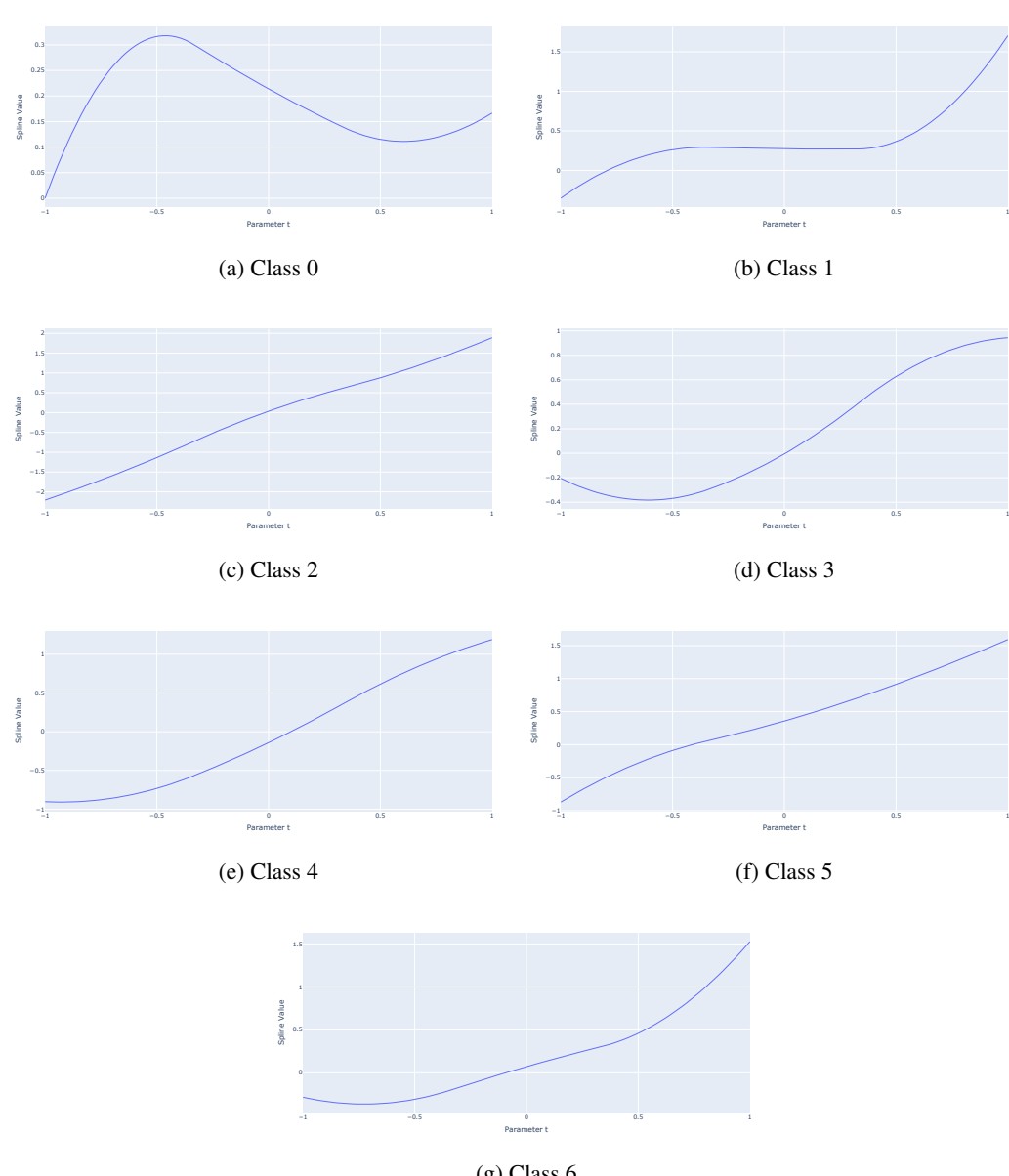

(a) Class 0

(b) Class 1

(c) Class 2

(d) Class 3

(e) Class 4

(f) Class 5

(g) Class 6

Figure 3: Visualization of the splines for the seven output neurons in the KANG model trained on the Cora dataset. Each output neuron has an associated matrix of spline weights with dimensions $D \times G$, where $D$ is the output dimension of the previous layer and $G$ corresponds to the grid coordinates. The rows of this matrix, which correspond to the contributions of each neuron in the previous layer, are summed to produce a vector of size $G$. This vector represents the aggregated spline weights that modulate the transformations applied by the neurons of the previous layer, providing insight into how features are combined in the final output, and can also be used to plot the corresponding splines.

## A.4 SCALABILITY STUDY

All the times reported represent the averages obtained from five training runs. Figures 4, 5, and 6 show how the models scale with increasing graph densities, with edge existence probabilities of 0.05, 0.25, and 0.5, respectively. In Figures 5b and 6b, excluding KANG, the times also account for the overhead of training GNNExplainer for the same number of epochs. The average epoch time for

GNNExplainer is then added to the average epoch time for each model. All scalability experiments were conducted on a g5.xlarge AWS instance.

We notice how KANG scales very well with increasing graph sizes and densities, with training times remaining relatively constant. In contrast, GAT's training times increase significantly with graph size and density, making it impractical to train on graphs with 20,000 nodes and a density of 0.5.

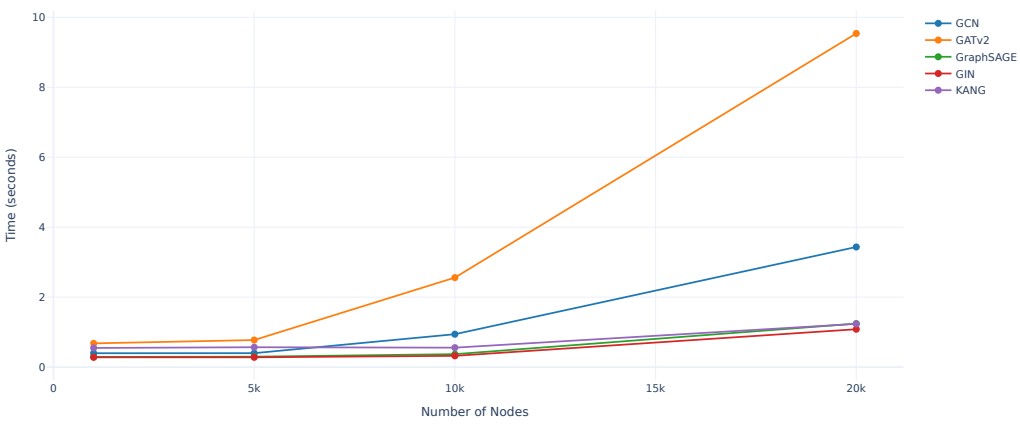

(a) Training Times

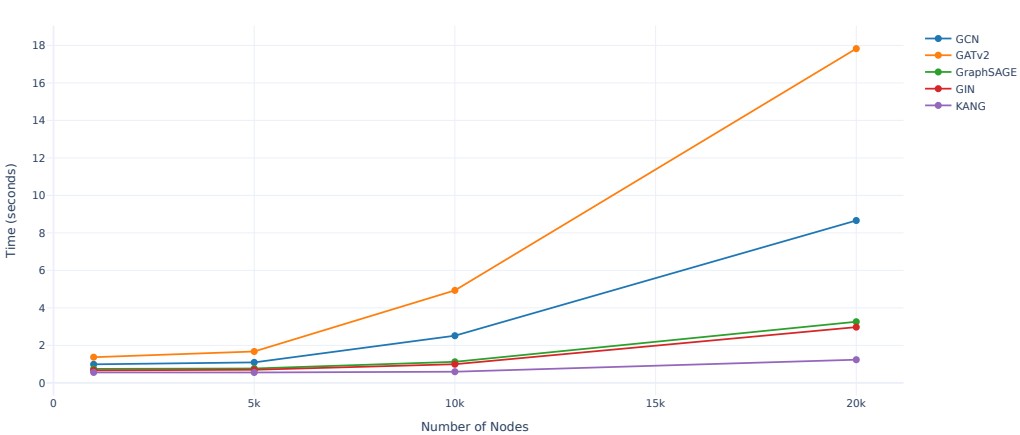

(b) Training and Explainability Times

Figure 4: The average training time per epoch is presented as a function of graph size, with the edge density fixed at 0.05. These times were obtained by averaging five training runs for each model and graph size.

Figure 7 illustrate how the models' inference times vary with different graph sizes and densities and Figure 8 shows the VRAM utulized by each model.

### A.5 MORE INTERPRETABILITY EXAMPLES

In this section, we compare the interpretability provided by KANG to the post-hoc explainability offered by GNNExplainer. Figure 9 illustrates the edge importance values generated by both methods

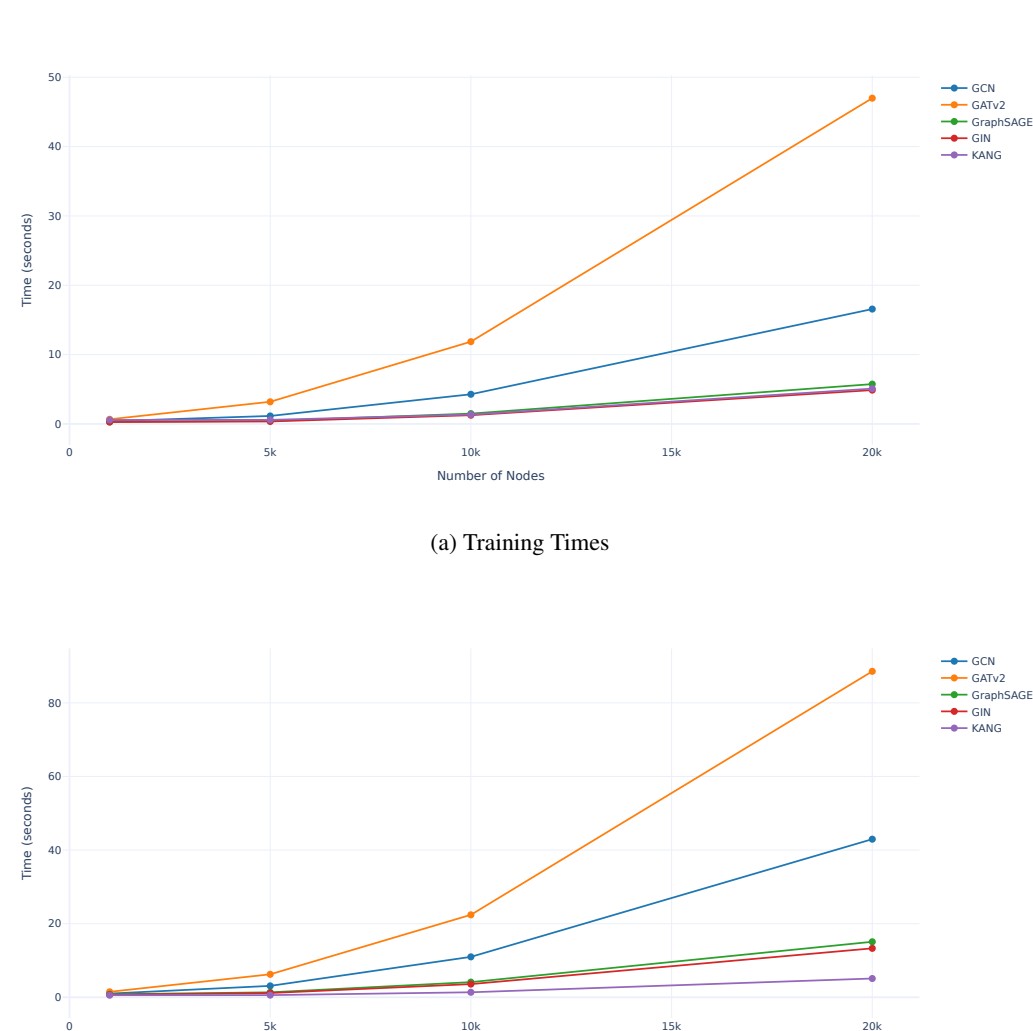

(a) Training Times

(b) Training and Explainability Times

Figure 5: Average epoch times for graphs with density 0.25.

for two target nodes (IDs 1761 and 1901) from the Cora dataset. The objective is to demonstrate how KANG's inherent interpretability, which directly computes edge importance using spline-based activations, aligns with the explainability produced by GNNExplainer. This comparison validates that the messages passed through the graph in KANG accurately represent the model's decision-making process without requiring additional post-hoc explainability methods.

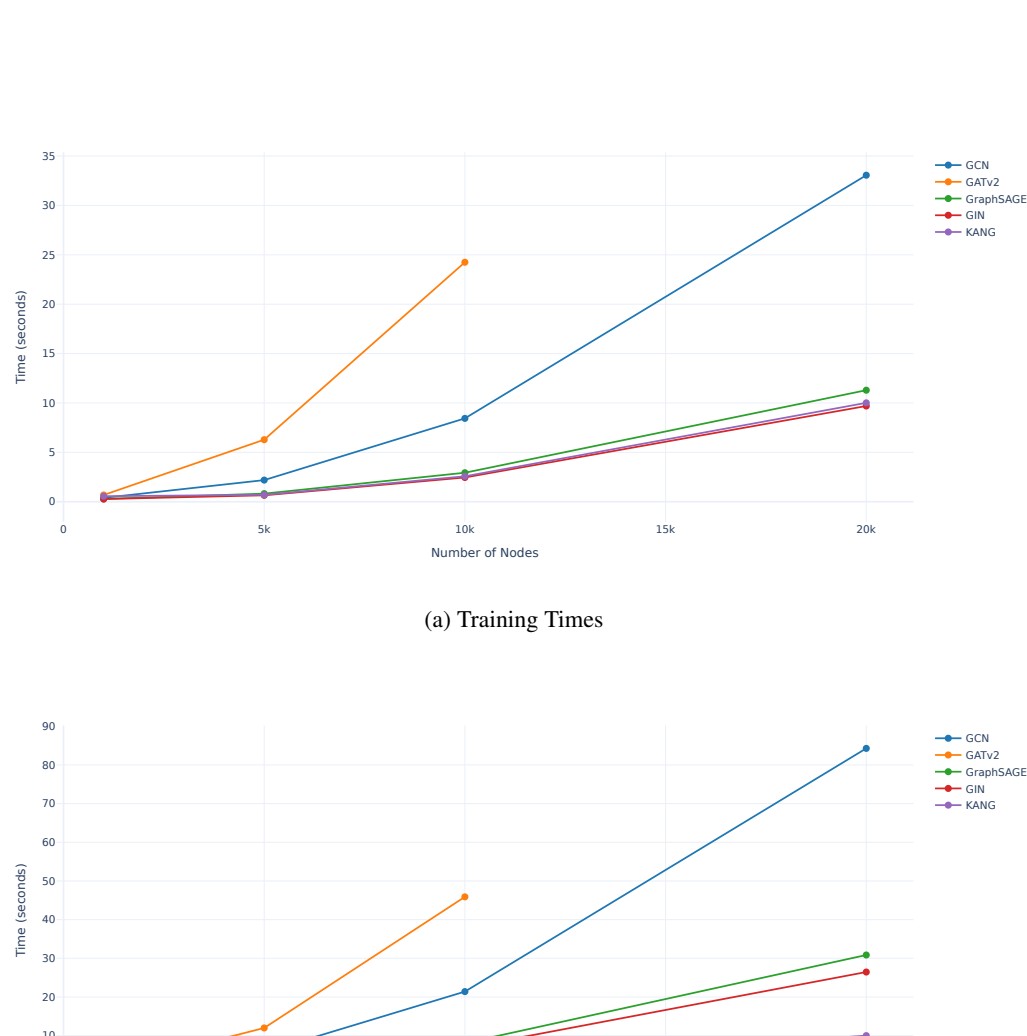

(a) Training Times

(b) Training and Explainability Times

Figure 6: Average epoch times for graphs with density 0.5. In Figure 6b are reported also the times for training GNNExplainer. For this graph density, GAT was unable to complete training on a graph with 20,000 nodes using an NVIDIA A10 with 24GB of VRAM.

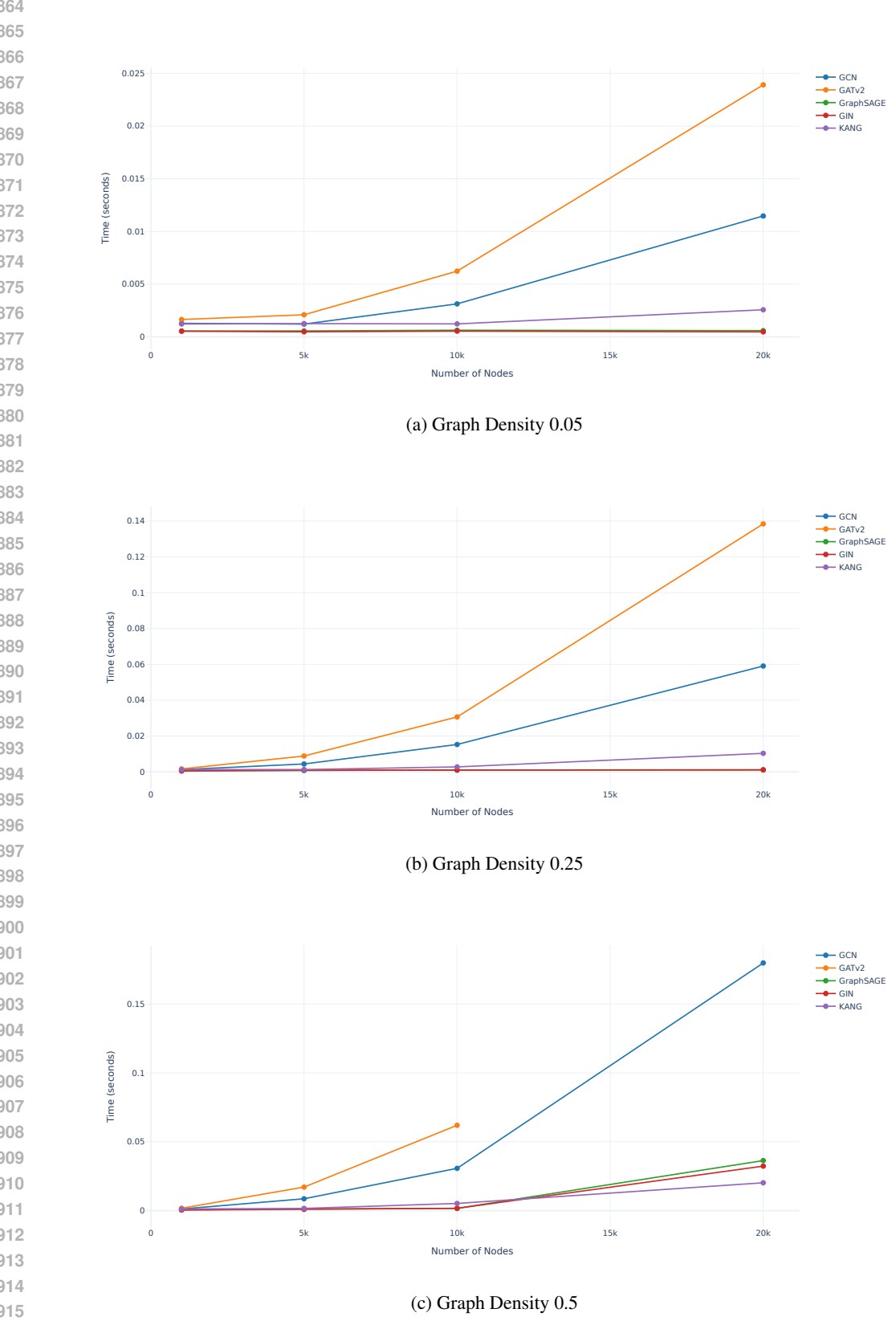

(a) Graph Density 0.05

(b) Graph Density 0.25

(c) Graph Density 0.5

Figure 7: Average inference times are shown in Figures 7a, 7b, and 7c, corresponding to graph densities of 0.05, 0.25, and 0.5, respectively. For the density of 0.5, we could not test the inference time of GAT as we were unable to complete its training on the available hardware.

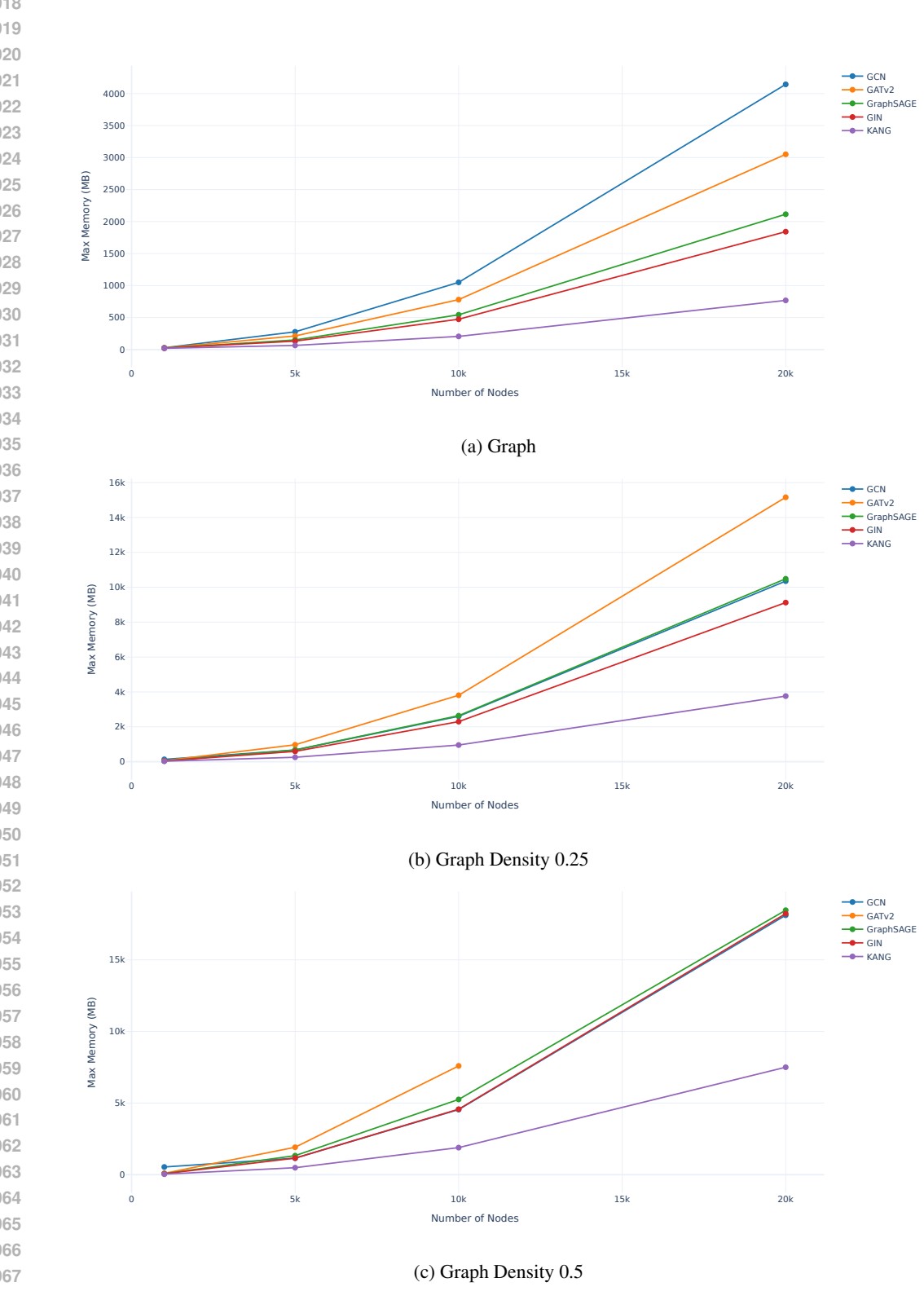

(a) Graph

(b) Graph Density 0.25

(c) Graph Density 0.5

Figure 8: Maximum GPU memory utilization for each graph size and density is shown in Figures 8a, 8b, and 8c, corresponding to graph densities of 0.05, 0.25, and 0.5, respectively.

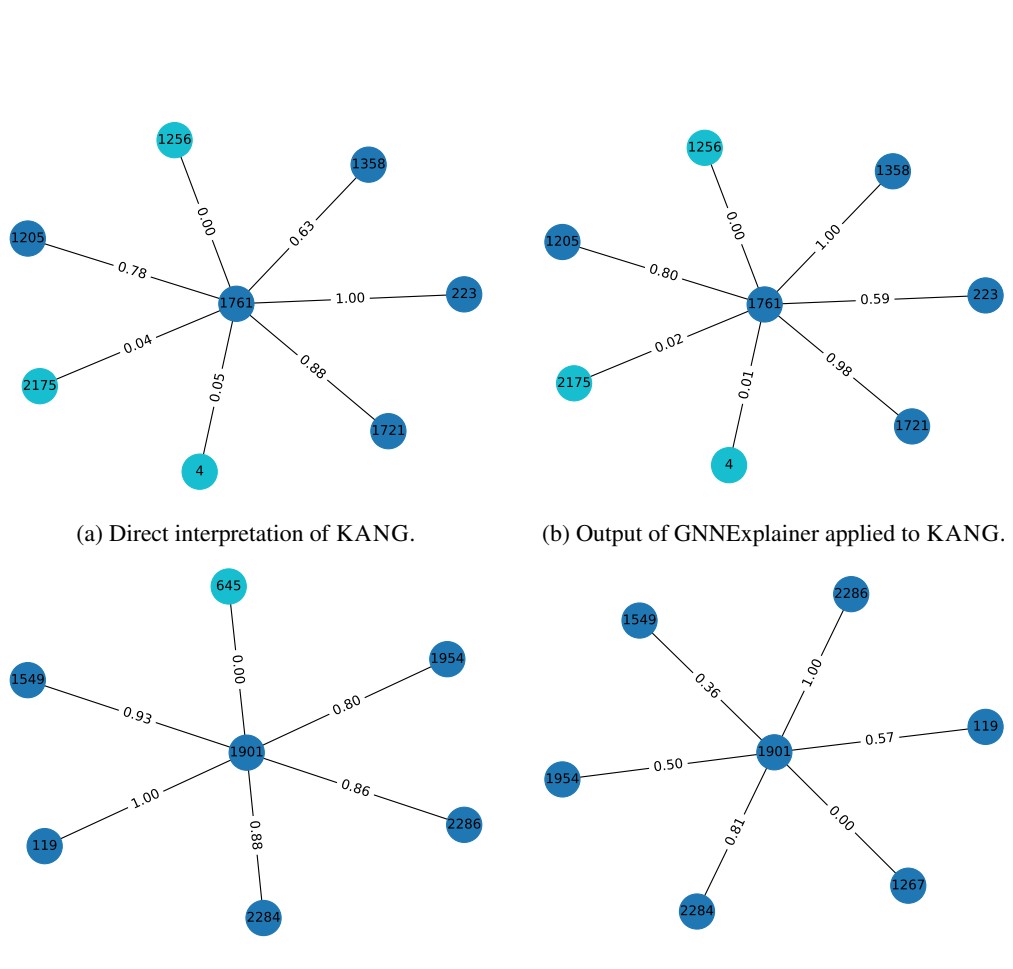

(a) Direct interpretation of KANG.

(b) Output of GNNExplainer applied to KANG.

(c) Direct interpretation of KANG.

(d) Output of GNNExplainer applied to KANG.

Figure 9: Interpretability vs. explainability. Figures 9a and 9c show the direct edge importance interpretation of KANG, illustrating the contribution of each neighboring node to the target node (1761 and 1901, respectively). The values represent the normalized edge importances derived from KANG's spline-based activations. Figures 9b and 9d depict the results from GNNExplainer applied to KANG, where edge importances are computed post-hoc. In both cases, KANG's direct interpretation aligns closely with GNNExplainer, validating the model's inherent interpretability.

