# OpenReview forum: "Kolmogorov–Arnold Graph Neural Networks"
_ICLR.cc/2025/Conference — Submitted to ICLR 2025_

### Official Review · Reviewer_dGuM · 2024-10-17

**Soundness:** 2
**Presentation:** 2
**Contribution:** 2
**Rating:** 3
**Confidence:** 3

**Summary:**

In this paper, the authors propose a novel graph neural network (GNN) model named KANG, which combines Kolmogorov–Arnold Networks (KANs) with GNNs and employs spline-based activation functions to enhance the accuracy and interpretability of GNNs. Different from existing methods, KANG can inherently provide interpretability, which is suitable for transparent decision-making. Experiments are conducted on several benchmark datasets, which demonstrates that KANG can outperform existing GNN methods in various graph tasks, such as node classification, link prediction, and graph classification.

**Strengths:**

- Different from traditional GNN methods, KANG can inherently provide interpretability for GNNs, which is particularly valuable for transparency. Additionally, KANG integrates interpretability directly into its architecture, which removes the need for additional post-hoc explainability tools.
- Moreover, the spline-based activation functions in KANG provide increased flexibility in modeling complex relationships in graph data, which helps capture nonlinear interactions more effectively and contributes to inprove the performance and interpretability of KANG.

**Weaknesses:**

- The writing of this paper could be improved. For instance, the introduction is overly long and lacks clarity, which makes it difficult for readers to understand the key motivations and contributions of the work.
- The motivation of this paper is unclear, which is challenging to understand the specific limitations of existing GNN methods. KANG emphasizes enhanced interpretability and performance, but it does not adequately highlight the practical issues or real-world scenarios.
- The novelty of KANG is limited, as it simply combines Kolmogorov–Arnold Networks (KANs) with GNNs. Although this integration enhances interpretability and performance of GNNs, it does not introduce fundamentally new mechanisms or theoretical advancements beyond those already present in KANs and existing GNN architectures.
- The experimental results are not convincing, and the baseline methods are not strong enough. Although the authors compare KANG with widely used GNN methods like GCN, GAT, GIN, and GraphSAGE, these methods are neither the latest nor the most competitive, which undermines the significance of the reported performance.

**Questions:**

- The authors should provide clearer motivations to explain why combining KAN and GNNs is necessary.
- The time and space complexity of KANG should also be provided in method part.
- More recent GNN baseline methods should be introduced in the experimental part for fair comparisons, such as rLap [1], UGNN [2], and PANDA [3] for graph classification, as well as GRLC [4], SEK [5] and UGT [6] for node classification.

[1] Kothapalli V. Randomized schur complement views for graph contrastive learning[C] International Conference on Machine Learning. PMLR, 2023: 17580-17614.

[2] Luo X, Zhao Y, Qin Y, et al. Towards semi-supervised universal graph classification[J]. IEEE Transactions on Knowledge and Data Engineering, 2023, 36(1): 416-428.

[3] Choi J, Park S, Wi H, et al. PANDA: Expanded Width-Aware Message Passing Beyond Rewiring[C] Forty-first International Conference on Machine Learning.

[4] Peng L, Mo Y, Xu J, et al. GRLC: Graph representation learning with constraints[J]. IEEE transactions on neural networks and learning systems, 2023.

[5] Yao T, Wang Y, Zhang K, et al. Improving the expressiveness of k-hop message-passing gnns by injecting contextualized substructure information[C] Proceedings of the 29th ACM SIGKDD Conference on Knowledge Discovery and Data Mining. 2023: 3070-3081.

[6] Lee O J. Transitivity-Preserving Graph Representation Learning for Bridging Local Connectivity and Role-Based Similarity[C] Proceedings of the AAAI Conference on Artificial Intelligence. 2024, 38(11): 12456-12465.

---

> ### Author Response · Authors · 2024-11-26
> **Response to reviewer dGuM**
>
> Dear Reviewer,
>
> We appreciate your time and effort in reviewing our paper and providing detailed feedback. Below, we address each of your concerns and questions, aiming to clarify misunderstandings and provide additional context where necessary.
>
> Reviewer Comment:
>
> The motivation of this paper is unclear, which is challenging to understand the specific limitations of existing GNN methods. KANG emphasizes enhanced interpretability and performance, but it does not adequately highlight the practical issues or real-world scenarios.
>
> Response:
>
> We believe the motivation behind our work has been clearly stated. However, we recognise the importance of presenting it with greater clarity. In future revisions, we will strive to communicate our message more effectively.
>
> Reviewer Comment:
>
> The novelty of KANG is limited, as it simply combines Kolmogorov–Arnold Networks (KANs) with GNNs. Although this integration enhances interpretability and performance of GNNs, it does not introduce fundamentally new mechanisms or theoretical advancements.
>
> Response:
>
> While the integration of Kolmogorov–Arnold Networks with GNNs may seem incremental, we believe that our spline-based activation functions represent a meaningful architectural innovation, offering improved adaptability and intrinsic interpretability.
>
> Reviewer Comment:
>
> The experimental results are not convincing, and the baseline methods are not strong enough. Although the authors compare KANG with widely used GNN methods like GCN, GAT, GIN, and GraphSAGE, these methods are neither the latest nor the most competitive.
>
> Response:
>
> We appreciate your suggestion to include more recent methods such as rLap, UGNN, PANDA for graph classification, and GRLC, SEK, and UGT for node classification. While we agree that including newer methods can enrich our comparisons, our choice of baselines, including GCN, GAT, GIN, and GraphSAGE, was deliberate. These architectures are well-established, widely used, and continue to serve as benchmarks. By comparing KANG against these methods, we aim to position our work in relation to the most commonly used GNNs, ensuring broader relevance. That said, we recognise the value of also including novel approaches.
>
> Reviewer Comment:
>
> The introduction is overly long and lacks clarity, which makes it difficult for readers to understand the key motivations and contributions of the work.
>
> Response:
>
> We acknowledge that the introduction could be more concise. We will streamline this section, focusing on clearly communicating the problem statement, motivations, and key contributions of KANG.
>
> Reviewer Comment:
>
> The time and space complexity of KANG should also be provided in the method part.
>
> Response:
>
> Thank you for pointing this out. We have provided a scalability study in Appendix A2. We agree that providing also a detailed analysis of time and space complexity can ensure a comprehensive understanding of KANG’s computational requirements.
>
> Thank you again for your thoughtful feedback. We will incorporate your suggestions in the future version of our work.
>
> Best regards,
>
> The authors

---

> > ### Comment · Reviewer_dGuM · 2024-11-26
> >
> > Thank you for your feedback. Some concerns still are not solved, such as Q3. I retain my score.

---

### Official Review · Reviewer_shva · 2024-10-17

**Soundness:** 1
**Presentation:** 4
**Contribution:** 2
**Rating:** 3
**Confidence:** 4

**Summary:**

This paper introduces KANG, a Graph Neural Network (GNN) model that incorporates Kolmogorov–Arnold Network Layers into the message-passing framework typical of GNNs. In KANG, messages from node $j$ are computed using spline-based activation functions in each layer. The paper claims that KANG 1) outperforms state-of-the-art GNN models on standard graph tasks (Cora, PubMed, Citeseer, MUTAG, and PROTEINS), and 2) provides interpretable results with negligible computational overhead.

**Strengths:**

1) The main strength is the focus on the explainability of the proposed Network. KANs are known to be, by construction, more interpretable than standard Neural Networks [1]. Extending this to both feature and edge influence with virtually no cost is interesting and well-explained.

2) The authors decided to include brief discussion of other KAN-based GNNs at the beginning of the paper even if they are new and still non-peer-reviewed. This paragraph provides helpful context.

3) The paper is clear and provides the code for reproducibility.

**Weaknesses:**

## Main Weaknesses
---

Having tested and re-run part of the provided code, I have several concerns regarding the empirical evaluation and some significant errors in the code:

1) I ran `comparison.py` to reproduce the Link prediction results for all models (GCN, GATv2, GraphSAGE, GIN, KANG) on the three datasets (Cora, PubMed and CiteSeer). My results differ significantly from those in Table 1. Here are the results I obtained:

| Dataset  | GCN        | GAT        | SAGE       | GIN        | KANG       |
|----------|------------|------------|------------|------------|------------|
| Cora     | 89.5 ± 0.7 | **90.5 ± 0.5** | 85.4 ± 2.4 | 80.8 ± 8.5 | 87.9 ± 0.6 |
| PubMed   | **94.2 ± 0.4** | 90.0 ± 0.2 | 86.4 ± 1.1 | 89.3 ± 0.5 | 85.2 ± 0.3 |
| CiteSeer | 85.3 ± 3.0 | **86.8 ± 2.6** | 82.5 ± 2.8 | 84.2 ± 1.0 | 85.9 ± 0.5 |

These results seem to contradict one of the two main claims of the paper:
"Our experiments [...] demonstrate that KANG outperforms state-of-the-art GNN models in node classification, link prediction, and graph classification tasks." (lines 13-15). I recommend the authors use more seeds (>= 30) and ensure that all runs converge (some results show high standard deviation, indicating potential convergence issues).

2) In the scalability experiment (lines 357–367), the number of parameters is fixed across networks. This comparison is not fair, as models like GCN tend to require fewer parameters to obtain optimal performances and thus may scale more efficiently. The comparison should be made at fixed performance levels. On my GPU (NVIDIA RTX A5000), KANG's training time was approximately 10 times slower than GCN for Cora, PubMed, and CiteSeer, contradicting the claim that KANG only "slightly" increases computational cost (line 363).

3) For node classification, models are validated on the test set and tested on the validation set. This makes it difficult to compare results with existing literature, where models like GCN and GATv1 might outperform KANG on tasks for Cora and CiteSeer [2], [3].

4) GAT models typically perform better with additional attention heads [3]. The hyperparameter for attention heads should be tuned (e.g., in the set [2, 4, 8]).

5) In the link predictor experiment, average training time per epoch is calculated as ((end_time - start_time) / n_epochs). This calculation is invalid if early stopping is triggered.

## Minor Weaknesses
---

1) KANG introduces three additional hyperparameters (spline grid size, spline degree, and aggregation function). For the hyperparameter search space you considered, this addition makes the hyperparameter search 126 times slower. This fact should be highlighted.

2) The statement "For each run, to achieve unbiased outcomes, we randomly split the datasets, utilizing 80% for training, 10% for validation, and 10% for testing" (lines 355–356) is not accurate for all experiments.

3) The `node_classification.py` script is located only in the `del` folder and throws an error when run both inside and outside the folder. I did not investigate why.

4) There are few typos in the paper:
- In Equation (2) `b()` is missing
- In lines 225-226 change "incluence" with "influence"

**Questions:**

1) What is the synthetic dataset used in the scalability study of Appendix A.4? I didn't see it explained neither in the main paper nor in the appendix. Its description should be included in the paper.

2) Inspired by GNNs, you experimented with `average()`, `sum()` and  `max()` aggregation functions for KANG. Since the Kolmogorov–Arnold representation theorem holds for the `sum()`, the other two aggregation functions do not have the same theoretical guarantees. However, in Table 4, Appendix A2 you actually show that `mean()` is favoured. Is this statistically significant, or is it not an important hyperparameter to tune? Did you investigate what happens to the splines when different aggregations are used? Do you think this result extends to deeper networks and/or traditional KAN?

---

> ### Author Response · Authors · 2024-11-26
> **Response to reviewer shva**
>
> Dear Reviewer,
>
> We sincerely thank you for your detailed feedback and for taking the time to reproduce part of our experiments. Below, we address your comments and provide responses to your concerns and questions.
>
> Comment:
>
> The empirical results differ significantly from Table 1. Results show high standard deviation, indicating potential convergence issues. The authors should use more seeds (>= 30) and ensure all runs converge.
>
> Response:
>
> We appreciate your effort in rerunning our code. It is concerning that your results differ from those reported in our paper. We checked our experiments and results and they seem consistent. However, we will rerun all experiments and we will investigate potential discrepancies in the experimental setup (e.g., random seeds, library versions, or GPU configurations) that may have caused the deviation in results. While we used 10 seeds in our experiments to balance computational resources available and statistical rigor, we agree that increasing the number of seeds could improve reliability.
>
> Comment:
>
> The scalability experiment comparison is not fair because the number of parameters is fixed across networks. Models like GCN require fewer parameters to obtain optimal performance. Additionally, KANG’s training time on Cora, PubMed, and CiteSeer was approximately 10 times slower than GCN on an NVIDIA RTX A5000 GPU.
>
> Response:
>
> Thank you for your comment. For the scalability study we weren't concerned by the performances of each individual model, but we were interested in how models with comparable dimensions scale to different sizes of data. In future work we will also add a study where we perform scalability experiments with fixed performance levels across models.
>
> Regarding training time, it would be helpful if you could share your specific experimental setup, together with the results you obtained. This will allow us to investigate any discrepancies more thoroughly and ensure reproducibility of results.
>
> We want to underline that, with the same number of parameters, KANG scales better compared to the methods it was benchmarked against. If you are referring to training times for node classification, it is possible that KANG may be slower due to its intrinsic need for a larger number of parameters to obtain higher classification performances.
>
> Comment:
>
> Models are validated on the test set and tested on the validation set, making comparisons with existing literature difficult.
>
> Response:
>
> We apologize for the oversight. Since all the models have been trained, tested, and validated on the same splits, we believe our results still hold. However, we will revise our code to align with existing literature.
>
> Hyperparameter Tuning
>
> Comment:
>
> GAT models typically perform better with additional attention heads. The hyperparameter for attention heads should be tuned (e.g., in the set [2, 4, 8]).
>
> Response:
>
> Thank you for the suggestion. In our initial experiments, we fixed the number of attention heads to 1 for consistency across baselines. We will tune the number of attention heads for GAT-based models within the suggested range.
>
> Comment:
>
> KANG introduces three additional hyperparameters, significantly increasing the hyperparameter search space. This should be highlighted.
>
> Response:
>
> As described in the manuscript, KANG has two additional hyperparameters (grid size and spline order). However we will emphasise this better when describing the hyperparameter optimisation in the future work.
>
> Code Reproducibility
>
> Comment:
>
> The node_classification.py script is located only in the del folder and throws an error when run both inside and outside the folder.
>
> Response:
>
> We appreciate your diligence in testing our code. The code in the "del" folder is legacy code that we will delete in the future version. The scripts to run are located in the root folder, like 'comparison.py'. There is an inconsistency in the naming provided in the readme, that we will correct in future releases.
>
> Minor Errors and Typos
>
> Comment:
> Equation (2) is missing $b()$, and there is a typo in lines 225–226 (“incluence” should be “influence”).
>
> Response:
> Thank you for catching these imprecisions. We will correct those.
>
> Response to Q1
>
> Thank you for your question. The synthetic dataset used in Appendix A.4 is a randomly generated graph dataset with controllable graph sizes and densities.
>
> Response to Q2:
>
> Thank you for this insightful question. To clarify, the aggregation function in KANG affects only how the node embeddings are aggregated to update the hidden state of each node during the message-passing process. It does not alter the theoretical foundation of the Kolmogorov–Arnold representation theorem or the way splines are summed. The representation theorem remains fully intact in KANG.
>
> We are grateful for your thorough review and constructive comments. Your feedback has been helpful in identifying areas for improvement, and we are committed to addressing all concerns.
>
> Best regards,
>
> The authors

---

> > ### Comment · Reviewer_shva · 2024-11-27
> >
> > Thank you for your response. I believe that addressing the points raised by other reviewers and me and emphasizing KANG's intrinsic explainability could result in a valuable paper. Best wishes for your future resubmission.

---

### Official Review · Reviewer_u76L · 2024-10-30

**Soundness:** 3
**Presentation:** 3
**Contribution:** 3
**Rating:** 3
**Confidence:** 4

**Summary:**

This paper presents a novel graph neural network (GNN) model KANG, built on the recently emerged Kolmogorov-Arnold network (KAN). KANG has a KAN-based convolution layer for the propagation and aggregation of messages between nodes, and a KAN-based linear layer to perform a linear transformation on the aggregated node features. The main advantage of KANG is its inherent interpretability (thanks to KAN), making it different from existing post-hoc GNN explainers. For this purpose, KANG computes feature influence (leveraging interaction of the gradients and the spline weights at each KAN hidden layer) and edge importance (combing spline weights and activations of the two node features), and use both of them to interpret the outcomes of KANG. Through extensive experiments on five benchmark datasets, KANG is demonstrated to outperform SOTA GNN models in node classification, link prediction, and graph classification tasks. In particular, feature influence of KANG is analysed on the CORA dataset, and edge importance is also shown to be consistent with the explainability provided by GNNexplainer using the same dataset.

**Strengths:**

This is a timely paper to introduce KAN for graph representation learning, with a focus on increasing interpretability of existing GNN models. This makes it different from other papers in integrating KAN and GNNs (see a few examples in the second last paragraph of Section 1). KANG utilises information flow across the graph and different KAN layers to develop two new notions, feature influence and edge importance, for its interpretability. I appreciate their effort to use insightful information from KAN to enhance interpretability. The evaluation in the paper shows that KANG has strong performance in both accuracy and interpretability for node classification, link prediction, and graph classification tasks. Most parts of the paper are clearly written, and some suggestions on how to further improve the paper can be found in the next review section.

**Weaknesses:**

Overall, I like the idea of KANG, but I also notice the following weaknesses.

1. Presentation of the paper can be still improved. I give a few examples.

1.1 Section 2.2: Figure 1 can be improved to better illustrate the main idea of KANG. I cannot fully understand why node x6 is connected to the two dots in the part of KANG convolution.

1.2 Section 2.2.2: ${\bf m}_j^{(l)}$ is defined, but not used in the following formulation. Should it be $x_j^{(l)}$? The notion $N(i)$ is used but not defined. If it is the set of direct neighbours of node $i$, the aggregation function at the KANG convolution layer does not take node $i$'s own feature into account.

1.3 Section 2.3.3: I don't fully understand the weights ${\bf W}^{(l)}$ and how to compute mean(${\bf W}^{(l)} \cdot S_{mean}^{(l)}$).

1.4 When reading the paper, I found a few typos: "is it possible", "at layer".

2. I believe the main contribution of KANG is about interpretability, but the evaluation of KANG's interpretability in this paper is insufficient.

2.1 The evaluation of feature influence and edge importance is only demonstrated on the CORA dataset.

2.2. KANG is only compared with GNNExplainer.

3.3 The paper claims that KANG interpretability is consistent with the explainability provided by GNNExplainer. Somehow, I am not fully convinced by this statement. For example, in Figure 2, node 2175 in Figure 2(a) is almost as important as node 2176, but in Figure 2(b), it has a much smaller importance than node 2176. Figure 9 in the Appendix has similar problems: node importances (values and ranks) are quite different for KANG's direct interpretation and GNNExplainer's explanation.

**Questions:**

1. I would encourage the authors to perform a comprehensive evaluation of KANG's interpretability, considering datasets used by existing post-hoc GNN explanation methods, and more baselines such as PGExplainer and EdgeSHAPer.

2. It is interesting to see that KANG outperforms SOTA GNN models (GCN, GAT, GraphSAGE, GIN) in node classification, link prediction, and graph classification tasks. Can you provide more information about the models, such as hidden dimension, the total number of parameters?

3. In general KAN networks are difficult to train. Appendix A.4 shows that KANG scales very well. This is a bit surprising. Can you give more explanation?

---

> ### Author Response · Authors · 2024-11-26
> **Response to reviewer u76L**
>
> Dear Reviewer,
>
> We sincerely thank you for your thorough and insightful review of our paper. Your feedback is invaluable in helping us improve the clarity and quality of our work.
>
> We are delighted that you find our integration of KAN into GNNs both timely and valuable. Your positive remarks about our efforts are highly encouraging.
>
> Below, we address each of your comments and questions in detail.
>
> **Presentation of the paper can still be improved**
>
> 1. Section 2.2: Figure 1 can be improved to better illustrate the main idea of KANG.
>
> Response:
>
> Thank you for pointing this out. We will revise Figure 1 to more accurately depict the KANG convolution process.
>
> 2. Section 2.2.2: $\textbf{m}_j^{(\ell)}$ is defined but not used in the following formulation. Should it be $x_j^{(\ell)}$ ? The notion $N(i)$ is used but not defined. If it is the set of direct neighbors of node $i$, the aggregation function at the KANG convolution layer does not take node $i$’s own feature into account.
>
> Response:
>
> Thank you for highlighting this. We will provide clearer formulas in the future versions.
>
> $N(i)$ is the set of direct neighbours of $i$, the aggregation takes into account also the hidden representation of node $i$. We will correct the formula accordingly.
>
> 3. Section 2.3.3: I don’t fully understand the weights $\textbf{W}^{(\ell)}$ and how to compute $mean(\textbf{W}^{(\ell)} \cdot S_{mean}^{(\ell)})$.
>
> Response:
>
> Thank you for pointing out the ambiguity in this section. We will expand Section 2.3.3 to provide a clearer explanation of how the weights are computed.
>
> 4. When reading the paper, I found a few typos: “is it possible”, “at layer”.
>
> Response:
>
> Thank you for noting these typos. We will correct them in the future version.
>
> **I believe the main contribution of KANG is about interpretability, but the evaluation of KANG’s interpretability in this paper is insufficient**
>
> 1. The evaluation of feature influence and edge importance is only demonstrated on the CORA dataset.
>
> Response:
>
> We agree that a broader evaluation would strengthen our claims. We will extend our interpretability analysis to include additional benchmark datasets commonly used in GNN explainability studies.
>
> 2. KANG is only compared with GNNExplainer.
>
> Response:
>
> You are correct that including more baseline methods would enhance the evaluation. We will incorporate comparisons with additional state-of-the-art post-hoc GNN explanation methods.
>
> 3. The paper claims that KANG interpretability is consistent with the explainability provided by GNNExplainer. Somehow, I am not fully convinced by this statement. For example, in Figure 2, node 2175 in Figure 2(a) is almost as important as node 2176, but in Figure 2(b), it has a much smaller importance than node 2176. Figure 9 in the Appendix has similar problems: node importances (values and ranks) are quite different for KANG’s direct interpretation and GNNExplainer’s explanation.
>
> Response:
>
> Thank you for this observation. We will revise and extend our analysis to provide a more detailed comparison between KANG’s direct interpretation and the results obtained from post-hoc models like GNNExplainer.
>
> Response to Q1
>
> We appreciate this recommendation and will expand our experiments accordingly. By including more datasets and comparing KANG with other XAI methodologies, we aim to provide a more comprehensive evaluation of its interpretability.
>
> Response to Q2
>
> Certainly. We will update the table in Appendix A2 to include the total number of parameters for KANG and the baseline models.
>
> Response to Q3
>
> We acknowledge that training KAN networks can be challenging due to their complexity, however, the efficiency of our implementation allows us to mitigate the intrinsic complexity. We are confident that our scalability experiments have been thoroughly conducted, but we will repeat them to double-check their validity.
>
> We genuinely appreciate your constructive feedback, which will help us enhance our paper.
>
> Thank you for your time and consideration.
>
> Best regards,
>
> The authors

---

> > ### Comment · Reviewer_u76L · 2024-11-27
> > **Thank you for the responses**
> >
> > I really appreciate the responses by the authors to ALL the reviews. I believe that integrating KAN to improve the interpretability of GNNs is still a valid idea. I encourage the authors to revise and improve their paper, and wish them good luck with future submissions.

---

### Official Review · Reviewer_yhFB · 2024-11-01

**Soundness:** 2
**Presentation:** 2
**Contribution:** 2
**Rating:** 3
**Confidence:** 5

**Summary:**

This paper introduces Kolmogorov-Arnold Networks (KANs) for enhancing graph neural networks. The core idea is to apply KANs to the aggregation process, replacing linear transformations with KANs.

**Strengths:**

1. Applying KANs to GNNs is intriguing, and the writing is clear. The core idea of the paper is easy to grasp.

2. The experimental results demonstrate the superior performance of KANs.

**Weaknesses:**

1. I don't understand why a widely used benchmark (https://paperswithcode.com/sota/node-classification-on-cora-with-public-split) was not employed for evaluation.

2. The idea lacks novelty.

**Questions:**

N/A

---

> ### Author Response · Authors · 2024-11-26
> **Response to reviewer yhFB**
>
> Dear Reviewer,
>
> Thank you for taking the time to review our submission and providing your feedback. While we appreciate your recognition of the clarity of our writing and the intriguing nature of applying Kolmogorov-Arnold Networks (KANs) to Graph Neural Networks (GNNs), we would like to address your concerns and provide additional context regarding the perceived weaknesses.
>
> - Regarding the Evaluation on Widely Used Benchmarks:
>
> We appreciate the emphasis on widely used benchmarks like the public split of Cora. While we did include the Cora dataset in our experiments, we opted for a different split. We believe the chosen split does not diminish the relevance of our results, however, we acknowledge the value of including results on commonly used splits for broader comparability and we will add these evaluations in future work.
>
> - On the Novelty of the Idea:
>
> While it is true that enhancing GNN aggregation mechanisms has been explored, the use of KANs introduces a novel way of incorporating learnable spline-based activation functions at the edges of the model, which fundamentally differs from prior approaches. This architecture provides both improved expressiveness and intrinsic interpretability, which are not commonly addressed simultaneously in the existing literature. We believe this contribution offers a meaningful advancement in the field, and we are happy to clarify this distinction further in our future version.
>
> - On the Rating and Overall Assessment:
>
> While we respect your evaluation, we would appreciate additional specifics regarding your concerns about soundness or presentation, as this would help us improve the manuscript. We are committed to refining our work based on constructive feedback and ensuring its value is effectively communicated in future submissions.
>
> We thank you again, and we hope this response clarifies the intentions and contributions of our work.
>
> Best regards,
>
> The authors

---

> > ### Comment · Reviewer_yhFB · 2024-11-28
> >
> > I have read the rebuttal and raised the score to 3. Thanks

---

### Official Review · Reviewer_WfiL · 2024-11-02

**Soundness:** 2
**Presentation:** 2
**Contribution:** 2
**Rating:** 5
**Confidence:** 4

**Summary:**

The paper introduces the Kolmogorov-Arnold Network for Graphs (KANG), a new Graph Neural Network (GNN) model that enhances interpretability and accuracy in tasks like node classification, link prediction, and graph classification. Utilizing spline-based activation functions on edges, KANG not only improves performance on benchmark datasets but also allows for better insight into decision-making processes, eliminating the need for external explainability techniques. This approach is especially beneficial in domains where understanding model decisions is crucial.

**Strengths:**

originality: medium
quality: medium
clarity: medium
significance: medium

**Weaknesses:**

1. It's better to split the second paragraph in the first section for readability.

2. The paper borrows too much content from the original KAN paper and makes incremental modification.

3. In section 2.2.2, where is $m_j^{(l)}$ in the following equations? It seems that this notation is defined but not used. Check out the consistency of the notations.

**Questions:**

1. "This kind of neural network has an activation function on the edges instead of nodes" What is the edges and nodes here? Are they the same as the edges and nodes in graph?

2. "KANG employs learnable spline-based activation functions on the edges of the graph" Is this similar to graph attention network or graph transformer? What is the difference and what is your advantage?

3. How does KANG perform on heterophilic datasets, especially on those challenging datasets identified in [1]? And how does KANG compare with some SOTA models instead of the baseline models, e.g. [2,3].

4. How does KANG compare with other GNN interpretability methods, e.g. [4]?




[1] The heterophilic graph learning handbook: Benchmarks, models, theoretical analysis, applications and challenges. arXiv preprint arXiv:2407.09618. 2024 Jul 12.

[2] Revisiting heterophily for graph neural networks. Advances in neural information processing systems. 2022 Dec 6;35:1362-75.

[3] Simplifying approach to node classification in graph neural networks[J]. Journal of Computational Science, 2022, 62: 101695.

[4] Miao S, Liu M, Li P. Interpretable and generalizable graph learning via stochastic attention mechanism. InInternational Conference on Machine Learning 2022 Jun 28 (pp. 15524-15543). PMLR.

---

> ### Author Response · Authors · 2024-11-26
> **Response to reviewer WfiL**
>
> Dear Reviewer,
>
> Thank you for your valuable feedback. We appreciate the time and effort you dedicated to reviewing our work.
> Below, we address each question in detail.
>
> Response to Q1
>
> Thank you for your insightful question. The terms “edges” and “nodes” in the context of KANG are indeed analogous to those in a graph, but with a conceptual adaptation tailored to the design of KANs.
> In KAN, the “nodes” represent the computational units where information is aggregated (neurons in traditional neural networks).
> The “edges” represent the connections between these nodes. However, unlike in traditional neural networks that are static, in KAN, and consequently also in KANG, the edges are equipped with activation functions. These activation functions are learnable and parameterized (e.g., splines) to allow non-linear transformations of the information as it flows between the nodes of the neural network.
> In summary, while the terms “nodes” and “edges” borrow their nomenclature from graph theory, in KANG they reflect the constituent units of the neural architecture.
> In the new version of the paper we try to better distinguish between nodes and edges of the graph and those of the neural network.
>
> Response to Q2
>
> Thank you for your follow-up question. To clarify, as explained previously, the learnable spline-based functions in KANG are not applied to the edges of the graph itself. Instead, these functions are associated with the edges of the neural network model, which serve as connections between computational units (nodes). These edges carry signals between nodes, and the learnable splines provide non-linear transformations of these signals, enhancing the expressiveness of the network.
> Advantages of KANG:
> - Fine-grained Non-linear Transformations: The spline-based functions allow each edge of the model to apply its own learnable non-linearity, enabling more nuanced transformations of data compared to fixed activation functions in traditional GNNs.
> - Adaptability and Interpretability: The splines are not only highly expressive but also interpretable, providing insights into the relationships modeled by the network.
>
> Response to Q3
>
> Thank you for raising this important question. We acknowledge the significance of evaluating models on heterophilic datasets and comparing their performance against state-of-the-art (SOTA) approaches.
> At this stage, we have not yet tested KANG on heterophilic datasets, including those challenging datasets identified in [1]. Evaluating KANG in such scenarios is indeed a valuable research direction that we aim to explore in future work.
>
> Regarding comparisons with SOTA models like those in [2,3], our current experiments focus on baseline models to establish the foundational performance and capabilities of KANG. Incorporating SOTA comparisons is another logical next step to fully contextualize KANG’s effectiveness and potential advantages.
> We greatly appreciate this suggestion and will prioritize both these aspects in follow-up work to better position KANG within the broader GNN research landscape.
>
> Response to Q4
>
> Thank you for your insightful question regarding the comparison of KANG with other GNN interpretability methods.
> To clarify, KANG is not designed as a post-hoc interpretability method. Instead, it is a Graph Neural Network that inherently incorporates interpretability into its design through the use of learnable spline-based activation functions.
>
> While KANG is not directly comparable to post-hoc explainability algorithms such as GNNExplainer, PGExplainer, or GCExplainer, we are currently conducting experiments to compare the interpretability of KANG with these methods. Specifically, we are investigating how KANG’s intrinsic interpretability aligns with or surpasses the results provided by these post-hoc approaches. This includes assessing fidelity, consistency, and clarity of the explanations generated by KANG versus those produced by external explainability frameworks.
> We believe that KANG’s design, by enabling built-in interpretability, offers a more robust foundation compared to relying solely on post-hoc methods, which may sometimes struggle to faithfully represent the underlying model’s decision-making process. We will present these comparisons and insights in future work as we further analyze KANG’s interpretability capabilities.
>
> Thank you once again for your constructive feedback.
>
> Best regards,
>
> The authors

---

> > ### Comment · Reviewer_WfiL · 2024-12-01
> > **Thanks for the reply**
> >
> > Thanks for the reply from the authors. After going through the comments from other reviewers, I will keep my rating.

---

### Author Response · Authors · 2024-11-26
**Comment to the Reviewers**

Dear Reviewers,

Thank you for taking the time to review our work and provide such detailed feedback. We acknowledge that the paper, in its current form, is not ready for publication.

That said, we greatly appreciate your constructive comments and suggestions, which have been incredibly helpful. We are committed to improving our work for future submissions by implementing your feedback and advice.

We hope that our responses have addressed your concerns, and helped to better communicate our contributions and motivations.

Best regards,

The Authors

---

### Meta-Review · Area_Chair_EA9D · 2024-12-19

**Metareview:**

This submission presents KANG, a novel Graph Neural Network (GNN) model that integrates Kolmogorov-Arnold Networks with GNNs and utilizes spline-based activation functions. The primary goal of KANG is to enhance the accuracy and interpretability of GNNs, enabling transparent decision-making processes. The authors evaluate their model on several benchmark datasets, highlighting its potential benefits.

Reviewers praised the clarity of the writing in some aspects, but also suggested improvements in presentation and notation. There was a consensus among reviewers that applying Kolmogorov-Arnold Networks to GNNs is an intriguing idea with significant potential, and KANG's inherent interpretability is a notable advantage, as it eliminates the need for external explainability techniques and makes it suitable for high-stakes domains.

However, the reviewers raised several concerns. Specifically, they noted that the computational overhead of KANG is a significant drawback, and the evaluation of its interpretability is insufficient. Additionally, some reviewers felt that the comparison to baseline models was inadequate, and there were technical issues with notation and formulas that need to be addressed.

To strengthen this submission, the authors should prioritize addressing these concerns, particularly by providing more comprehensive evaluations of KANG's interpretability, improving computational efficiency, and enhancing the clarity and rigor of their presentation. With revisions that address these issues, KANG has the potential to make a meaningful contribution to the field of GNNs.

**Additional Comments On Reviewer Discussion:**

The discusion between reviewers and authors were in line with the concerns reviewers had. The rebuttal did not help much.

---

### Decision · Program_Chairs · 2025-01-22

Reject